# Bioaccumulation Behavior and Human Health Risk of Polybrominated Diphenyl Ethers in a Freshwater Food Web of Typical Shallow Lake, Yangtze River Delta

**DOI:** 10.3390/ijerph20032671

**Published:** 2023-02-02

**Authors:** Bei Li, Juanheng Wang, Guocheng Hu, Xiaolin Liu, Yunjiang Yu, Dan Cai, Ping Ding, Xin Li, Lijuan Zhang, Chongdan Xiang

**Affiliations:** 1South China Institute of Environmental Sciences, Ministry of Ecology and Environment, The Postgraduate Training Base of Jinzhou Medical University, Guangzhou 510530, China; 2State Environmental Protection Key Laboratory of Environmental Pollution Health Risk Assessment, South China Institute of Environmental Sciences, Ministry of Ecology and Environment, Guangzhou 510530, China; 3College of Resources and Environment, Yangtze University, Wuhan 430100, China

**Keywords:** polybrominated diphenyl ethers (PBDEs), bioaccumulation, biomagnification, health risk, shallow lakes

## Abstract

Background: Polybrominated diphenyl ethers (PBDEs) have been commonly found in aquatic ecosystems. Many studies have elucidated the bioaccumulation and biomagnification of PBDEs in seas and lakes, yet few have comprehensively evaluated the bioaccumulation, biomagnification, and health risks of PBDEs in shallow lakes, and there is still limited knowledge of the overall effects of biomagnification and the health risks to aquatic organisms. Methods: In this study, a total of 154 samples of wild aquatic organism and environmental samples were collected from typical shallow lakes located in the Yangtze River Delta in January 2020. The concentrations of PBDEs were determined by an Agilent 7890 gas chromatograph coupled and an Agilent 5795 mass spectrometer (GC/MS) and the bioaccumulation behavior of PBDEs was evaluated in 23 aquatic organisms collected from typical shallow lakes of the Yangtze River Delta. Furthermore, their effects on human health were evaluated by the estimated daily intake (EDI), noncarcinogenic risk, and carcinogenic risk. Results: The concentrations of ΣPBDE (defined as the sum of BDE-28, -47, -100, -99, -153, -154, -183, and -209) in biota samples ranged from 2.36 to 85.81 ng/g lipid weight. BDE-209, BDE-153 and BDE-47 were the major PBDE congeners. The factors affecting the concentration of PBDEs in aquatic organisms included dietary habits, species, and the metabolic debromination ability of the PBDE congeners. BDE-209 and BDE-47 were the strongest bioaccumulative PBDE congeners in aquatic organisms. Additionally, except for BDE-99, BDE-153 and BDE-154, the trophic magnification factor (TMF) values of PBDE congeners were significantly higher than 1. Moreover, the log Kow played a significant role in the biomagnification ability of PBDE congeners. The noncarcinogenic risk of PBDE congeners and carcinogenic risk of BDE-209 from aquatic products were lower than the thresholds. Conclusions: PBDE congeners were bioaccumulated and biomagnified to varying degrees in aquatic organisms from typical shallow lakes. Both the noncarcinogenic and carcinogenic risks assessment of edible aquatic products indicated that none of the PBDE congeners pose health risks to the localite. This study will provide a basis for a comprehensive assessment of PBDEs in aquatic ecosystems in shallow lakes and for environmental prevention measures for decision-makers.

## 1. Introduction

Brominated flame retardants (BFRs) are the most widely produced and used flame retardants in the world. BFRs are used in electronics, textiles and other commodities in the form of additives and reactants, and additive-type flame retardants have low costs and high flame-retardant efficiency [1]. Polybrominated diphenyl ethers (PBDEs) are a class of synthetic aromatic chemical containing bromine atoms and were once the most important BFRs worldwide [2]. The technical PBDE products include three commercial mixtures, i.e., Penta-BDE (BDE-47, BDE-100, BDE-99, BDE-153, BDE-154), Octa-BDE (BDE-153, BDE-183, BDE-209) and Deca-BDE (BDE-209) [3,4]. Although the Stockholm Convention on Persistent Organic Pollutants regulated the production and usage of Penta-BDE and Octa-BDE in 2009 and Deca-BDE in 2017, a significant number of products on the market still contain PBDE congeners. In addition, Deca-BDE is still widely used in China and numerous other Asian nations [5,6]. The environmental risks of PDBEs are of great concern due to their heavy use in the last few decades and their environmental persistence [7]. At present, PDBEs are still widely detected in water, sediment, dust, and other environmental media. Zhou, et al. [8] detected eight congeners of PBDEs in the dust of the e-waste industrial parks and adjacent residential houses in south China, ranging from 3.158 × 10^−3^ to 15.648 × 10^−3^ mg/kg, and Wang, et al. [9] detected seven congeners of PBDEs ranging from 1.62 × 10^−3^ to 23.1 × 10^−3^ mg/kg in river sediments from Qingdao.

PBDEs have hydrophobic and highly lipophilic properties (Log Kow 5.98–9.97), which allows them to accumulate in the food web, biomagnify, and potentially harm aquatic ecosystems [10,11,12]. Once released into the aquatic ecosystems, PBDEs accumulate in primary producers and thus biomagnify through the food web to senior consumers. Fish play an important role in the food web and are often consumed by humans, which leads to the potential impacts of PBDEs in aquatic ecosystems on human health. Understanding the bioaccumulation and biomagnification of PBDEs is important to understand the ecological and environmental risks of hydrophobic and persistent chemicals. Choo, et al. [13] used GC-HRMS to determine the PBDEs content of 20 species in the marine food web of southern Korea and analyzed the distribution characteristics of PBDEs; the results showed that PBDEs suggesting strong bioaccumulation and biomagnification in marine ecosystems. Furthermore, Zhou, et al. [14] and Windsor, et al. [15] identified PBDEs with different levels of bioaccumulation and biomagnification in river and aquatic lake ecosystems. In addition, several animal toxicological studies and human epidemiological investigations have shown that PBDEs are cytotoxic and carcinogenic to fish and mammals, and can cause neurodevelopmental disorders, thyroid hormone disorders, and cancer in humans [16,17,18]. However, there have been few studies on human health risks caused by the bioaccumulation and biomagnification of PBDEs in aquatic lake ecosystems. This topic clearly requires more research in order to understand the biomagnification capacity and potential health risks of PBDEs in lake ecosystems [19].

Ge Lake is the second largest lake in southern Jiangsu Province and is an important part of the Taihu Lake basin of the Yangtze River Delta. Not only does it support flood discharge, water supply, and shipping but it also supports a biological habitat and fishing resources [20]. Changdang Lake is located in eastern Ge Lake and crosses the southwestern Hutang Textile Industrial Park in Changzhou and is mainly responsible for the supply of domestic water, the development of aquaculture, flood storage, and farmland irrigation, among other functions [21]. The Taihu Basin, located south of the Yangtze River Delta, is one of the most-developed and dynamic areas in China [22]. As an important part of Taihu Basin, Changdang Lake and Ge Lake are surrounded by developed industries. Pollution has increased significantly in Ge Lake and Changdang Lake for the past few years, with the development of the textile industry and electronics production in the surrounding area, and the continuous economic development and the acceleration of urbanization, which has seriously affected the water supply of normal residents and the healthy socioeconomic development of the Taihu Basin [23,24]. These freshwater ecosystems have many pollution sources, and aquatic organisms are a diverse and complete food web [25,26,27]. Therefore, bioaccumulation and biomagnification of aquatic food web can be used to study the potential health impacts on human beings, which make them good models for the in-depth study of the impacts to aquatic ecosystems in typical shallow lakes in the Yangtze River Delta region.

Previous studies have discussed the effects of nutritive salt, heavy metals, and concentrations of BFRs on the Taihu Lake basin [28,29,30], but there is little comprehensive research on bioaccumulation, biomagnification, and the health risks of PBDEs in shallow lakes, and there is still limited knowledge of the overall effect of biomagnification and the health risks to aquatic organisms. Therefore, this paper targeted the freshwater ecosystems of typical shallow lakes in the Yangtze River Delta. The research objectives were: (1) to assess the PBDEs pollution levels in aquatic organisms in typical shallow lakes; (2) to appraise the bioaccumulation and biomagnification potential of aquatic organisms in typical shallow lakes; and (3) to assess the noncarcinogenic risk and carcinogenic risk for the localite caused by the ingestion of aquatic products. The results will provide a thorough understanding of the bioaccumulation, biomagnification, and health risks in aquatic ecosystems of shallow lakes and will provide environmental prevention measures to decision-makers.

## 2. Materials and Methods

### 2.1. Materials and Reagents

PBDE standard mixtures (BDE-28, BDE-47, BDE-99, BDE-100, BDE-153, BDE-154, BDE-183, BDE-209) and BDE-77, BDE-118, BDE-128, BDE-205, ^13^C-BDE-209 were purchased from Cambridge Isotope Laboratories Inc. (Tewksbury, MA, USA), these standards all have a concentration of 50 μg/mL. The OASIS HLB SPE cartridge (500 mg/mL) was acquired from Waters Corporation (Milford, MA, USA). N-Hexane, dichloromethane, acetone, anhydrous sodium sulfate (Na_2_SO_4_), concentrated sulfuric acid (H_2_SO_4_), and sodium hydroxide (NaOH) were obtained from Anpel Laboratory Technologies (Shanghai) Inc. (Shanghai, China). Diatomaceous earth and quartz sand were purchased from BUCHI Labortechnik AG (Flawil, Switzerland). Water and silica gel were obtained from Millipore Corporation (Burlington, MA, USA). Aluminum oxide was purchased from Sigma Aldrich (St. Louis, MO, USA).

### 2.2. Sample Collection

Ge Lake (31°29′~31°41′ N, 119°45′~119°52′ E) covers a total area of 166.7 square kilometers. The average depth of the lake is 1.3 m. Changdang Lake (31°34′~31°39′ N, 119°30′~119°32′ E) covers a total area of 86.7 square kilometers, and the average depth of the lake is 1.1~1.2 m. A total of 154 samples of wild aquatic organism (plankton, crustaceans, zoobenthos and fishes) and environmental samples (sediment and surface water) were collected from typical shallow lakes (Ge Lake and Changdang Lake) located in the Yangtze River Delta in January 2020. The locations of these typical shallow lakes are illustrated in Appendix A. At least two fish of each type were collected, and the specific quantity was determined according to the site conditions. Approximately 1500 g of river snail and shrimp were collected, and the samples were divided into four parts. Zooplankton samples were collected from surface waters using a 0.112 mm trawl. Details of the biota samples are provided in the Appendix A. Sediments were collected using a grab sampler at a depth of 0.1 m below the sediment surface. For surface water collection, the mouth of the sampler was aligned with the direction of the water flow to collect the water and to separate and remove the settling solids. The collected samples were transported to the lab on ice and stored at −80 °C before analysis. Details of the environmental samples are provided in the Appendix A.

### 2.3. Experiment Analysis

The pretreatment method for biota samples is similar to that used in a previous study by Liu, et al. [31], with minor modifications. In brief, the whole bodies of zooplankton, the muscles of fish and the soft tissues of crab, shrimp and river snail were freeze-dried. After drying, 1.0 g of the sample was weighed and put into a glass tube. BDE-77, BDE-205 and ^13^C-BDE-209 were added as surrogate standards and extracted using accelerated solvent with 200 mL dichloromethane: hexane (1:1, *v*/*v*). The extract was evaporated to 1–2 mL and reconstituted in 10 mL hexane. The lipid content was determined gravimetrically from 1 mL of the extract. The remaining extract was prepared to 1.0 mL and purified with a multilayer silica gel column. The eluents were concentrated to approximately 100 µL. BDE-118 and BDE-128 internal standards were added, followed by instrumental analysis.

The sediment sample was passed through a stainless-steel sieve after freeze-drying, and a copper sheet was added to the 1.0 g sample. The subsequent treatment was consistent with that of the biota sample. The surface water sample pretreatment method is similar to that of Gu, et al. [32]. Briefly, filtered water (5 L) was extracted to collect the grain phase. Surrogate standards were added to the samples (BDE-77, BDE-205 and ^13^C-BDE-209) after freeze-drying. Solvents were extracted from all matrices using an SPE system and adhered to an HLB SPE column (Waters, 6 cc, 200 mg). Then, 70 mL dichloromethane:hexane:methanol (1:2:1, *v*/*v*/*v*) mixed solvent was added for elution. The subsequent treatment was consistent with that of the biota sample.

The concentrations of PBDEs were determined by an Agilent 7890 gas chromatograph coupled and an Agilent 5795 mass spectrometer. Details of relevant instrument analysis, stable isotope analysis, organic carbon analysis, and quality control are provided in the Appendix A.

### 2.4. Bioaccumulation and Trophic Magnification Factors

Bioaccumulation factors (BAFs) include the bioconcentration factor (*BCF*) and biota-sediment accumulation factor (*BSAF*). *BCF* refers to the ratio of the contaminant concentration detected in aquatic organisms and the dissolved water concentration, which represents the ability of aquatic organisms to absorb pollutants directly from the water, as shown in the equation [13]:$$BCF=C_{biota}/C_{water}$$
where *C_biota_* is the target compound concentration in the target organism (pg/kg lw), and *C_water_* is the concentration of the target compound in surface water (pg/L). When the target compound *BCF* value is greater than 5000 (or Log*BCF* > 3.7), target compound can be considered to bioaccumulate in the food web. When the *BCF* value is between 2000 and 5000 (or 3.3 < Log*BCF* < 3.7), the compound may potentially bioaccumulate.

*BSAF* refers to the ratio of the lipid base concentration in aquatic organisms to the organic carbon-based concentration in sediments, as shown in Equations [13]:$$BSAF=C_{biota}/C_{sediment}$$
where *C_biota_* is the target compound concentration in the target organism (ng/g lw) and *C_sediment_* is the concentration measured in organic matter and sediments (ng/g toc). When BSAF is greater than 1, there is a sediment bioaccumulation effect. When *BSAF* is less than 1, there is a sediment-bioaccumulated dilution effect.

The trophic levels (*TL*) of organisms relative to zooplankton were determined, which supposed occupying *TL* 2. For each individual sample of zooplankton, the *TL*s of aquatic organisms were determined using the following equations [33,34].
$$TL_{consumer}=\left(\delta^{15}N_{consumer}-\delta^{15}N_{zooplankton}\right)/3.4+2$$

Based on the regression equation between the concentrations of the target organism and *TL*s, the trophic magnification factor (*TMF*) value was determined as shown in equations [33,34]:lnConcentration=aTL+b
$$TMF=exp^{a}$$
where *a* is the empirical slope and *b* is the y-intercept. Based on the *p* = 0.05 significance level, when the *TMF* value was greater than 1, the analyte was considered to have biomagnification potential.

### 2.5. Human Health Risk Assessment

Dietary intake is a common route of human exposure to PBDEs, so dietary exposure to pollutants is the key to evaluating any health risks to the population. The following equations were used to determine the estimated daily intake (*EDI*) of PBDEs from the consumption of aquatic products [35]:$$EDI=\left(Concentration\times C_{food}\right)/BW$$
where *EDI* is measured in ng/kg bw/day, Concentration is the total PBDEs concentration (ng/g ww), *C_food_* is the daily consumption of aquatic products: adults 44.9 (g/day) [36], and BW is the adult consumers’ weight (63 kg) [37].

As reported by Yu, et al. [38], potential health risks, including noncarcinogenic risk and carcinogenic risk, were estimated by the dietary intake of PBDEs. As shown in the equations:HQ=EDI/RfD
HI=ΣHQ
CR=EDI×CSF
where *RfD* and *CSF* are the reference dose level and carcinogenic slope factor, respectively. The *RfD* and *CSF* values are provided in Appendix A. The thresholds for noncarcinogenic and carcinogenic risks are 1 and 1 × 10^−4^, respectively.

### 2.6. Statistical Analysis

IBM SPSS Statistics26 (IBM, Armonk, NY, USA) was used for statistical analysis. The data were nonnormally distributed according to the Shapiro–Wilk test. Therefore, Kruskal–Wallis nonparametric tests were used to analyze the difference in PBDEs among aquatic organisms. Spearman’s rank correlation coefficient was used to evaluate the correlation of bioaccumulation among the congeners of pollutants., and α = 0.05 was considered significant.

## 3. Results and Discussion

### 3.1. Concentrations and Congener Profiles of PBDEs

Table 1 shows the concentrations of PBDE congeners in 23 aquatic species. PBDEs were detected in all aquatic organisms and environmental samples, indicating that PBDEs are ubiquitous in aquatic organisms from the investigated typical shallow lakes. Except for BDE-99 and BDE-209, the detection frequencies (DFs) of all 6 PBDE congeners were >90%. The total concentrations of PBDE congeners in biota samples ranged from 2.36 (loach) to 85.8 ng/g lipid weight (lw) (Culter alburnus). In order from highest to lowest, the median concentrations were: Culter alburnus > River snail > Silver sillago > Common carp > Yellow catfish > Oriental sheatfish > Northern snakehead > Finless eel > Crab > Coilia nasus > Crucian carp > Shrimp > White semiknife carp > Aucha perch > Silver carp > Japanese eel > Plankton > Grass crap > White amur bream > Black carp > Loach > Bighead carp > English perch. The ∑_8_PBDE levels (2.36–85.81 ng/g lw) observed in this study were comparable to those in aquatic organism samples collected from Taihu Lake (8.6–74.3 ng/g lw) [39], while they were lower than those collected from Gaobeidian Lake (13.6–355 ng/g lw) and even lower than those in Baiyangdian Lake (160.2 ng/g lw) [34,40]. However, the median values of ∑_8_PBDEs in silver carp (9.08 ng/g lw), grass carp (7.4 ng/g lw), common carp (46.6 ng/g lw), crucian carp (17.9 ng/g lw) and yellow catfish (33.1 ng/g lw) were much higher than those in the corresponding species in Dianshan Lake: silver carp (4.3 ng/g lw), grass carp (0.26 ng/g lw), common carp (10 ng/g lw), crucian carp (5.8 ng/g lw) and yellow catfish (16 ng/g lw) [14]. Compared with the concentrations of PBDEs found in those species in fresh water in other studies in China, the aquatic biota in these typical shallow lakes had only medium concentrations of PBDEs. In addition, the ∑_8_PBDE concentrations were 2.70–5.95 ng/g TOC (mean 4.69 ng/g TOC) in the sediment samples and 165.28–297.59 pg/L (mean 231.44 pg/L) in the surface water samples in the present study.

Biological parameters and diet may play important roles in regulating the bioaccumulation of PBDEs in aquatic organisms. Carnivores had the highest concentration of PBDEs, excluding BDE-153 and BDE-183, followed by omnivores, and then herbivores (Figure 1A). This study found significant bioaccumulation of PBDEs in high TLs. The concentrations of ∑_8_PBDEs in carnivores and omnivores were significantly higher than in herbivores (*p* < 0.05), and the concentrations of BDE-28, BDE-47 and BDE-100 were significantly lower in omnivores than in carnivores (*p* < 0.05). In an aquatic ecosystem, carnivores typically occupy higher TLs and may accumulate higher concentrations of PBDEs when consuming other contaminated organisms. Omnivores mainly feed on algae, aquatic plants and zoobenthos, and had lower concentrations of PBDEs. Herbivores occupy the lowest TLs and mainly eat algae, duckweed, and other aquatic plants. Conversely, when comparing different species of aquatic organisms (Figure 1B), there were no significant differences in concentration for the congeners BDE-28, BDE-99 and BDE-209 in fish, crustacean and zoobenthic samples taken from typical shallow lakes (*p* > 0.05). However, the concentrations of BDE-47 in fishes were significantly higher than those in crustaceans and zoobenthos (*p* < 0.05). Studies have indicated that BDE-99 can be debrominated to form BDE-47 in common carp [14]. These results indicated that BDE-99 was metabolized strongly in these species, and perhaps led to the high concentration of BDE-47 in fishes. The concentrations of BDE-153 in crustaceans were also significantly higher than those in the other two taxa (*p* < 0.05). The crustaceans in this study were mainly shrimp and crab, which was consistent with the observation BDE-153 was the congener with the highest concentration in shrimp and crab in the Netherlands [41].

The profiles of PBDE congeners in twenty-three aquatic organisms in typical shallow lakes are depicted in Figure 2. BDE-209, BDE-153, and BDE-47 were the major congeners, accounting for 5–50% (mean: 21.3%), 7–38% (mean: 18.5%) and 3–41% (mean: 17.3%) of Σ_8_PBDEs, respectively, in aquatic biota excluding plankton. BDE-99 only contributed to 2.65% of Σ_8_PBDEs. The results of the BDE-209 analysis were similar to those of the predominant congener in the Yellow River and Pearl River sediments [42,43]. Since the United Nations banned the production and usage of commercial Penta- and Octa-BDE in 2009, China has begun to widely use Deca-BDE, mainly BDE-209. This is also consistent with the results that BDE-209 was the predominant congener detected by Wang, Jia, Gao, Zeng, Li, Zhou, Sheng, and Yu [29] in the sediments of Taihu Lake, indicating that Deca-BDE is the primary commercial product in the Taihu Lake basin. In addition, Luo, et al. [44] found that BDE-209 had a lower metabolic velocity, and that BDE-99 had one of the fastest metabolic velocities of PBDEs in aquatic organisms. This was probably why BDE-209 was a predominant congener but BDE-99 was not. However, Zeng, et al. [45] found that common carp could debrominate BDE-99 to BDE-47 and debrominate BDE-183 to BDE-154, and that BDE-209 could also be debrominated and metabolized into a low-brominated congener, so while the concentration of BDE-209 in common carp was low, BDE-47 and BDE-153 were the dominant congeners. A number of studies have demonstrated that debromination in fish is species-specific and affected by fat content and diet [46]. Additionally, this study found a similar pattern in the concentrations of BDE-47 in aquatic organisms to the general pattern of BDE-47 dominance in aquatic species [47]. One study showed that highly brominated congeners were debrominated into BDE-99 and then BDE-47; therefore, the metabolism of highly brominated PBDE congeners in aquatic organisms may be responsible for this result [48]. Interestingly, aquatic organisms tended to preferentially accumulate BDE-153 in addition to BDE-47 and BDE-209. Similar results were found in 17 freshwater fishes from the Yangtze River [49]. Since BDE-153 is an indicator of commercial Hexa-BDE, the higher proportion detected may be related to the use of consumer goods containing commercial Hexa-BDE.

BDE-209 was the major PBDE congener found in river snails and loach, which may be related to their benthic habitats and the presence of sediment or particulate matter in their digestive system [47]. Crucian carp and northern snakehead prefer bottom-dwelling activities and predation, and both accumulate BDE-209 through the ingestion of sediment-associated biota. BDE-183 was the dominant congener in plankton, accounting for 63% of congeners. Bradley, et al. [50] and He, et al. [51] found that Hepta- and Octa-BDE could be produced through microbial reductive debromination of BDE-209, that plankton included a large number of microorganisms, and that the low metabolic ability of plankton may inhibit the continued debromination speed of BDE183, explaining why the concentration of BDE-183 was high in plankton.

### 3.2. Bioaccumulation and Trophic Magnification Factors

BSAF and BCF have become common parameters used to evaluate the bioaccumulation ability of contaminants in aquatic organisms. Many studies have reported that PBDEs can bioaccumulate in aquatic biota along the food web [52,53]. We calculated the median BSAF and BCF values of PBDEs in this study (Table 2). BCF values higher than 5000 L/kg and BSAF values higher than 1 were found for all PBDE congeners in aquatic organisms in typical shallow lakes, indicating that PBDEs are highly bioaccumulative. This result was also similar to a study of biota in southern South Korea [13].

BDE-28 and BDE-153 showed higher contribution in aquatic organisms but not in sediments; therefore, BSAF of BDE-28 and BDE-153 had the two highest BSAFs of all the congeners, but BDE-183 was quite the opposite. Moreover, BDE-99 showed a lower contribution in aquatic organisms, and BDE-209 showed a higher contribution in sediments; thus, their BSAFs were lower than those of other PBDE congeners. The correlations between the BCFs of the different PBDE congeners were determined by Spearman’s rank correlation coefficient and are shown graphically in Figure 3. Different color intensities are used to represent different R values, ranging from 0 to 1, with darker colors showing stronger correlations. In general, there were strong correlations between BDE-154, BDE-99, BDE-47 and BDE-28 (r > 0.6, *p* < 0.05) and between BDE-100, BDE-47 and BDE-28 (r > 0.6, *p* < 0.05). These results implied that the metabolic debromination pathways are BDE-154 to BDE-99 to BDE-47 to BDE-28, and BDE-100 to BDE-47 to BDE-28. These results also confirmed the conclusions of previous studies [54,55,56]. The BCF values were significantly correlated among all PBDE congeners (*p* < 0.05), except between BDE-153 and BDE-209 (r < 0.15, *p* > 0.05), indicating that the bioaccumulation capacities of PBDEs in the study area were similar. Overall, based on the BCF and BSAF results in this study, BDE-28, BDE-47, BDE-153 and BDE-209 were the strongest bioaccumulative PBDE congeners in aquatic organisms. Low molecular weight PBDE congeners may be one factor leading to genetic recombination in mammalian cells, and less-brominated PBDE congeners were more likely to have negative effects on thyroid hormones than more-brominated PBDE congeners [57]. Therefore, the accumulation of BDE-28 and BDE-47 in aquatic organisms deserves attention.

The TMFs were used to evaluate the food web biomagnification potential of the entire lake ecosystem. Generally, food web biomagnification was determined by a statistical TMF greater than 1. Previous studies have pointed to higher PBDE concentrations in organisms with higher TLs than in their prey, suggesting that PBDEs could be biomagnified along the food web [58]. The TLs were listed in the rank order zooplankton (primary consumers) < herbivorous fish < omnivorous and carnivorous fish. In this study, except for BDE-99, BDE-153 and BDE-154, the plot of natural log lipid weight concentrations of PBDE congeners of aquatic organisms in typical shallow lakes was significantly positively correlated with TL (*p* < 0.05; slope > 0), and the TMF values of PBDE congeners were significantly higher than 1 (Figure 4). These results indicated that PBDE congeners could biomagnify in aquatic food webs in typical shallow lakes. The greatest TMFs were observed for BDE-28 (TMFs = 1.65, *p* < 0.05), whereas those of BDE-154 (TMFs = 0.9, *p* > 0.05) were not significantly different from 1, indicating that BDE-154 was not biomagnified across the food web. Table 2 shows the statistical results of the regression analysis. The calculated TMF for BDE-28 in typical shallow lakes (1.65) was higher than that of the freshwater food web in Baiyangdian Lake, China (1.41) [34], but was close to that in Lake Ontario fishes in Canada (1.60) [59]. Factors such as environmental conditions (e.g., temperature and distribution), PBDE concentrations in the food web, physicochemical properties of PBDE congeners (e.g., hydrophobic property, Kow), and species differences in food web components may all influence food web biomagnification and contribute to this phenomenon [60].

The Kow is an important attribute that determines the environmental fate of hydrophobic organic chemicals, especially in biota. The Kows of PBDEs were found to be associated with the biomagnification of PBDEs [61]. Appendix A shows the relationship between the TMFs of PBDE congeners and their log Kows. The results showed that the TMF values of PBDE congeners were negatively correlated with log Kow but tended to increase when log Kow exceeded approximately 8, excluding BDE-154 (TMF < 1). Shao, Tao, Wang, Jia and Li [58] showed that the biomagnification potential of hydrophobic organic pollutants with log Kows > 7 generally decreased with increasing Kow value in aquatic biota, and Tao, Zhang, Wu, Wu, Liu, Zeng, Luo and Mai [61] proposed that there was a significant negative correlation between the biomagnification potential of PBDEs and log Kow. In this study, with increasing hydrophobicity, the speed of PBDE congener uptake slowed, but the TMF increase with Kow for BDE-153 and BDE-183 was probably a result of their increased uptake efficiency and in vivo biodegradation. Obviously, log Kow was not the only physiochemical factor affecting the biomagnification potentials of PBDEs. Other factors, such as the molecular sizes of the chemicals, probably played a role [39].

### 3.3. Dietary Exposure and Risk Assessment

Many studies have demonstrated that one of the most important routes of human exposure to PBDEs is through dietary intake, and aquatic products are a major dietary component [62]. Appendix A shows the estimated daily intake (EDI) of PBDE congeners by the adult inhabitants of the study area through the consumption of aquatic products (except plankton). BDE-209, has the largest contribution to total dietary intake of the PBDE congeners, with an EDI value of 0.0194 ng/kg bw/day, followed by BDE-47 (0.0189 ng/kg bw/day) and BDE-153 (0.0147 ng/kg bw/day). Comparatively, the EDI value of BDE-99 was 0.0025 ng/kg bw/day, which was relatively low. It was also evident that common carp and river snail were major sources of human dietary exposure to PBDEs through aquatic products.

The noncarcinogenic risks of PBDEs were 40.7% in zoobenthos and 35.1% in fishes, followed by crustaceans (24.1%). Specifically, oriental sheatfish had the highest noncarcinogenic risk among PBDEs in all aquatic products, with an HI value of 0.0013. Because only the carcinogenic slope factor of BDE-209 was provided in the U.S. EPA documentation, this study was performed based on the carcinogenic risk assessment of BDE-209 for aquatic organisms in typical shallow lakes. The consumption of zoobenthos accounted for 60.2% of the carcinogenic risks for BDE-209, and crustaceans (24.9%) were slightly higher than fishes (14.9%). The relatively few species of zoobenthos in this study may be responsible for this result. Specifically, river snails (5.84 × 10^−11^) had the highest carcinogenic risk, and finless eels (4.38 × 10^−12^) had the lowest carcinogenic risk for BDE-209 in all aquatic products. Overall, the HI and CR values of aquatic organisms in typical shallow lakes were much lower than the established thresholds (Figure 5).

## 4. Conclusions

In this study, the levels of PBDEs in aquatic organisms from typical shallow lakes were used to analyze the congener profile, bioaccumulation and biomagnification potential, and human health risks of PBDE exposure through dietary intake. Compared with other studies, the contamination levels of these PBDE congeners were in the low to medium range. The relative contributions and distribution patterns of PBDE congeners in freshwater biota were different due to the biotas’ different diets, origins, biological parameters, metabolic ability, and physicochemical properties. Among the measured PBDEs, BDE-209 had the highest BCF, BDE-153 had the highest BSAF and BDE-28 had the highest TMF. The concentrations of most PBDE congeners were significantly positively correlated with trophic levels (*p* < 0.05), which indicated that PBDE congeners were bioaccumulated and biomagnified to varying degrees in aquatic organisms from typical shallow lakes. In addition, both the noncarcinogenic and carcinogenic risks of edible aquatic products were well below the thresholds, indicating that none of the PBDE congeners pose health risks to the localite. Based on these results, we can confirm that close attention to PBDE pollution in aquatic ecosystems is needed; although it does not pose a risk to human health, it is still of concern in the future. The shortcoming of this study is that the sample size is small. Since the samples are only from the Taihu Basin, the results of this study only represent shallow lakes in the Taihu Basin. Future studies should collect more extensive samples, pay attention to the pollution situation nationwide, and analyze the specific reasons affecting the bioaccumulation of PBDEs in organisms by studying the bioaccumulation characteristics of PBDEs in different biological tissues and edible parts of human beings, so as to provide a reliable scientific basis for the prevention and control of PBDE pollution.

## Figures and Tables

**Figure 1 ijerph-20-02671-f001:**
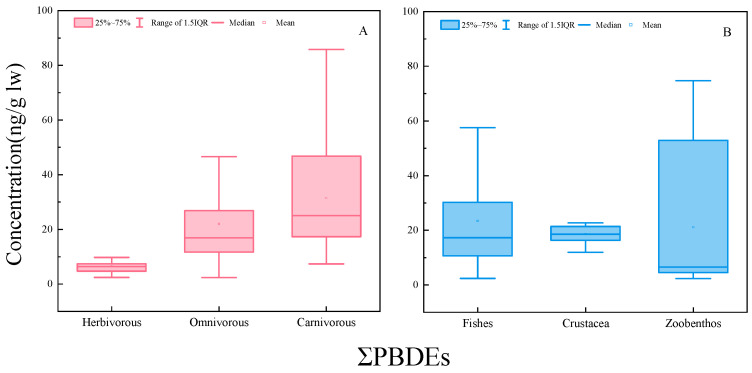
Concentration range of PBDE congeners in aquatic organisms of different species and different dietary habits collected from typical shallow lakes ((**A**) shows the concentration of PBDEs in aquatic organisms with different diet, while (**B**) shows the concentration of PBDEs in aquatic organisms with different species).

**Figure 2 ijerph-20-02671-f002:**
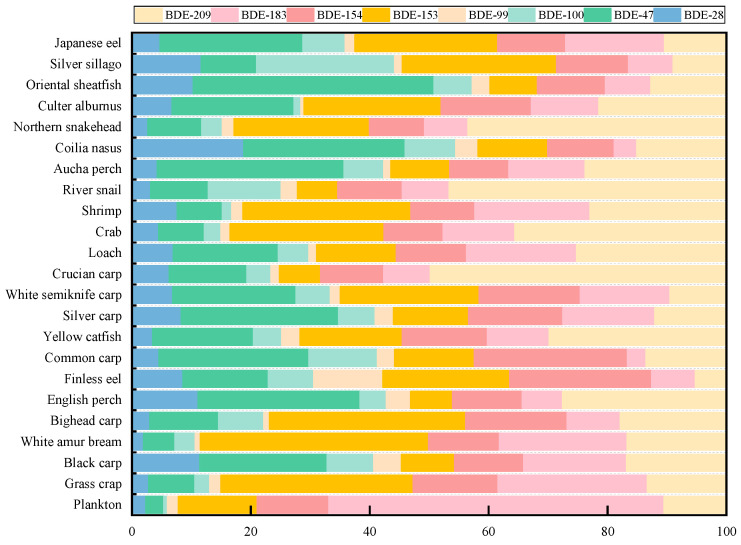
Composition profiles of PBDEs in aquatic organisms collected from typical shallow lakes.

**Figure 3 ijerph-20-02671-f003:**
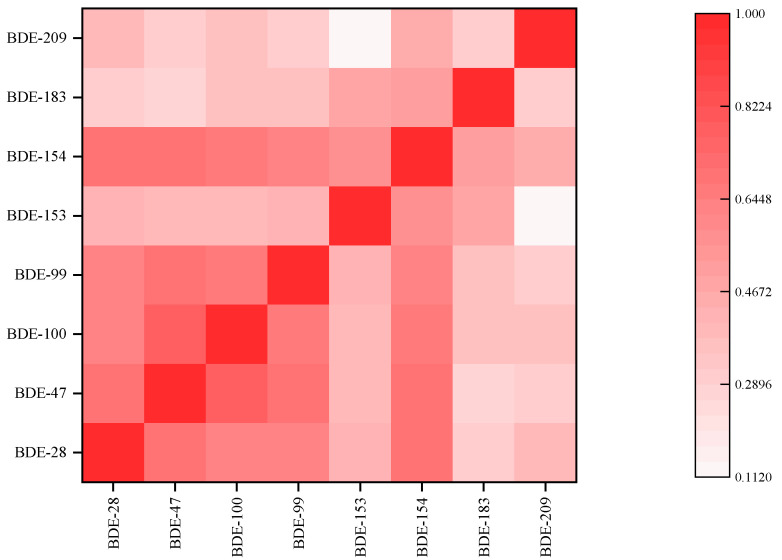
The correlations between the BCF in different PBDE congeners in aquatic organisms collected from typical shallow lakes.

**Figure 4 ijerph-20-02671-f004:**
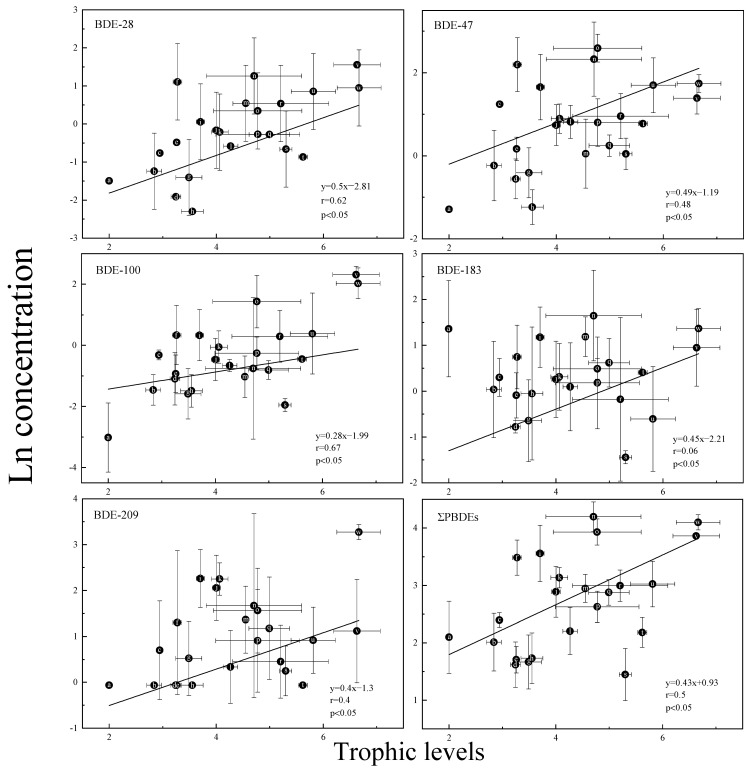
Relationship of the natural logarithm of PBDE congener concentration (ng/g lw) and TLs calculated from δ^15^N values (*p* < 0.05). (In the figure: a: Plankton; b: Grass crap; c: Aucha perch; d: Bighead carp; e: Black carp; f: Oriental sheatfish; g: Loach; h: White amur bream; i: Yellow catfish; j: Crucian carp; k: Northern snakehead; l: Silver carp; m: Shrimp; n: Culter alburnus; o: Common carp; p: White semiknife carp; q: Crab; r: Finless eel; s: English perch; t: Japanese eel; u: Coilia nasus; v: Silver sillago; w: River snail.).

**Figure 5 ijerph-20-02671-f005:**
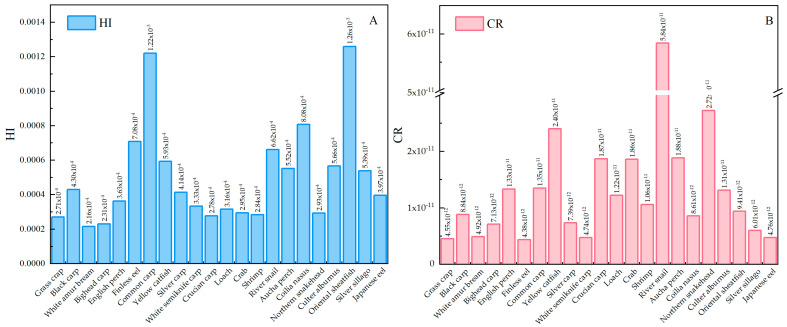
Noncarcinogenic and carcinogenic risks of PBDEs in aquatic organisms collected from typical shallow lakes ((**A**) shows the noncarcinogenic risk values of PBDEs, and (**B**) shows the carcinogenic risk values of BDE-209).

**Table 1 ijerph-20-02671-t001:** Concentrations of PBDE congeners (ng/g lipid weight) in aquatic organisms from typical shallow lakes in January 2020.

Sample	BDE-28	BDE-47	BDE-100	BDE-99	BDE-153	BDE-154	BDE-183	BDE-209	∑8PBDE
DF	99.3%	100%	98.6%	80.1%	100%	100%	97.3%	59.6%	-
Plankton	0.21 ^a^	0.27	0.06	0.16	1.19	1.08	5.05	0.94	8.97
	0.21 ^b^	0.27	0.06	0.16	1.19	1.08	5.05	0.94	8.97
	0.2–0.25 ^c^	0.27–0.28	nd–0.11	nd–0.27	0.97–1.4	0.88–1.27	1.86–8.23	nd	5.23–12.7
Grass crap	0.34	1.02	0.26	0.15	3.07	0.97	1.37	0.94	8.12
	0.2	0.55	0.18	0.13	2.29	1.01	1.78	0.94	7.4
	0.18–0.65	0.43–2.08	0.17–0.42	nd–0.27	1.72–5.19	0.62–1.27	0.31–2.03	nd	4.54–12.4
Black carp	0.64	1.2	0.44	0.26	0.5	0.65	0.97	0.94	5.59
	0.64	1.2	0.44	0.26	0.5	0.65	0.97	0.94	5.59
	0.49–0.78	0.96–1.44	0.26–0.61	0.23–0.28	0.15–0.85	0.36–0.94	0.64–1.29	nd	4.69–6.49
White amur bream	0.11	0.31	0.25	0.10	1.98	0.65	1.74	0.94	6.10
	0.11	0.3	0.19	0.05	2.16	0.67	1.21	0.94	6.39
	nd–0.18	0.13–0.46	0.1–0.43	nd–0.47	0.15–3.4	0.17–0.98	nd–4.63	nd	2.43–9.75
Bighead carp	0.16	0.61	0.40	0.05	1.74	0.9	0.47	0.94	5.26
	0.16	0.61	0.40	0.05	1.74	0.9	0.47	0.94	5.26
	0.1–0.21	0.41–0.8	0.18–0.61	nd	1.31–2.16	0.4–1.4	0.42–0.51	nd	3.82–6.69
English perch	0.57	1.1	0.14	0.11	0.25	0.74	0.24	1.42	4.57
	0.38	0.93	0.15	0.14	0.24	0.4	0.23	0.94	3.38
	0.37–0.96	0.77–1.59	0.11–0.17	nd–0.15	0.23–0.29	0.4–1.41	0.21–0.27	nd–2.39	3.16–7.18
Finless eel	2.29	2.91	1.80	2.13	3.48	3.95	2.00	2.09	20.6
	1.55	2.59	1.38	2.1	3.85	4.31	1.32	0.94	19.8
	0.52–5.34	1.13–5.07	0.47–4.54	0.23–3.63	0.26–6	0.6–6.43	nd–6.55	nd–4.43	15–26.9
Common carp	2.18	14	5.34	1.72	7.17	14	1.66	6.01	52
	2.54	14	6.42	1.6	7.45	14.3	1.72	7.52	46.6
	0.11–3.27	8.5–21	1.26–9.45	0.7–2.77	4.5–9.51	6.33–22.2	1.19–2.02	nd–8.29	40.2–66
Yellow catfish	1.23	6.67	1.88	1.22	6.40	6.35	3.92	11.4	39.1
	1.16	5.64	1.58	1.03	5.71	4.77	3.45	9.91	33.1
	0.23–2.54	0.99–17.8	0.49–5.68	0.11–2.84	0.45–17.8	1.38–14.4	1.32–8.86	3.35–22.4	14.2–72.2
Silver carp	0.6	2.41	0.53	0.22	1.17	1.37	1.50	1.85	9.65
	0.65	2.07	0.48	0.24	0.99	1.24	1.21	0.94	9.08
	0.29–0.8	1.55–3.96	0.45–0.7	nd–0.35	0.58–2.11	0.74–2.27	0.32–3.24	nd–4.59	5.58–14.9
White semiknife carp	0.83	2.67	0.88	0.18	2.25	1.62	1.71	4.16	14.3
	0.68	2.05	0.57	0.17	2.31	1.68	1.49	0.94	13.2
	0.38–1.75	1.32–8.34	0.43–1.91	nd–0.36	0.94–4.94	0.1–3.22	0.21–4.75	nd–9.81	8.06–22.6
Crucian carp	0.99	2.36	0.76	0.41	1.88	1.69	1.73	9.79	19.6
	0.98	2.03	0.63	0.22	1.08	1.65	1.22	7.74	17.9
	0.19–2.56	1.03–5.81	0.09–1.72	nd–1.99	0.37–5.08	0.36–3.43	0.27–4.34	2.34–21.7	7.01–35
Loach	0.32	0.79	0.29	0.1	0.85	0.51	0.66	2.36	5.88
	0.26	0.66	0.19	0.05	0.5	0.44	0.69	0.94	5.28
	0.07–0.86	0.33–2.2	0.06–1.15	nd–0.26	0.09–3.03	0.09–1.34	nd–1.15	nd–8.5	2.36–12.4
Crab	0.79	1.32	0.46	0.27	5.62	2.41	2.09	5.17	18
	0.75	1.29	0.46	0.26	4.32	1.66	2.01	5.94	18.8
	0.47–1.14	0.88–2.01	0.24–0.68	nd–0.61	1.91–11.9	1.21–6.12	0.81–3.55	nd–12.3	11.9–22.7
Shrimp	1.95	1.36	0.44	0.57	4.99	1.99	3.55	4.70	19.6
	1.38	1.38	0.29	0.34	5.12	1.95	3.51	4.16	17.2
	0.95–3.48	0.31–2.54	0.19–1.15	0.17–1.81	1.59–6.78	1.46–2.57	2–5.58	nd–9.54	14–29.3
River snail	3.91	5.81	8.52	2.08	3.09	6.40	4.22	26.7	60.7
	1.79	5.41	6.88	1.51	3.8	6.07	4.42	26.1	58.5
	1.05–10.6	4.52–7.74	4.59–17.6	0.53–5.39	0.96–3.82	2.84–9.82	2.19–6.76	21.3–32.4	52.9–74.8
Aucha perch	0.47	3.46	0.74	0.13	1.09	1.1	1.41	2.62	11
	0.47	3.46	0.74	0.13	1.09	1.1	1.41	2.62	11
	0.45–0.48	3.36–3.56	0.65–0.82	0.13–0.13	0.98–1.2	0.88–1.32	1.01–1.8	nd–4.3	10–12
Coilia nasus	3.2	6.55	2.91	0.81	2.44	2.41	0.77	3.07	22.1
	3.49	5.02	1.58	0.69	2.17	2.07	0.7	2.79	18.1
	0.34–6.73	1.78–13.2	0.18–10.4	0.22–1.81	0.51–4.91	1.26–4.13	nd–1.4	nd–6.19	12.9–39.5
Northern snakehead	0.94	2.59	1.07	0.51	4.28	2.23	1.67	10	23.3
	0.62	2.12	0.8	0.45	5.30	2.15	1.69	10.1	22.8
	0.45–1.74	1.77–4.29	0.59–2.07	0.22–0.97	0.34–7.35	1.52–3.39	0.5–2.78	5.77–15.4	17.3–29.9
Culter alburnus	5.65	13.25	1.62	0.41	14.23	10.6	6.80	15.6	68.2
	4.3	13.03	0.72	0.34	14.65	9.64	7.22	13.6	68.7
	0.86–13.2	3.34–23.6	nd–5.02	nd–0.89	9.61–18	5.52–17.7	1.24–11.5	nd–34.4	49.6–85.8
Oriental sheatfish	3.39	10.5	1.80	1.01	2.64	3.29	2.51	8.88	34
	2.93	11.5	1.82	0.84	2.27	3.24	2.16	3.61	28.8
	1.56–5.88	3.97–17.8	0.21–2.77	0.25–2	0.6–5.72	0.43–7.01	0.64–4.37	nd–25.8	24–49.3
Silver sillago	4.83	4.21	10.3	0.52	10.5	9.65	3.18	4.41	47.6
	4.5	3.56	8.94	0.5	9.98	4.64	2.89	3.44	48.3
	3.85–6.14	2.95–6.13	8.25–13.8	0.43–0.64	7.58–13.9	3.55–20.8	1.06–5.6	nd–8.84	45.6–48.9
Japanese eel	0.43	2.17	0.64	0.15	2.17	1.03	1.5	0.94	9.02
	0.43	2.17	0.64	0.15	2.17	1.03	1.5	0.94	9.02
	0.37–0.48	2.11–2.22	0.63–0.64	0.14–0.15	0.4–3.93	1.01–1.05	1.48–1.52	nd	7.37–10.7

nd: no detected; ^a^ mean; ^b^ median; ^c^ min-max.

**Table 2 ijerph-20-02671-t002:** BCF, BSAF, log Kow, slope and *p*-value of the slope of the regression analysis between the natural logarithm of concentration and TLs, and TMFs for PBDE congeners in aquatic organisms collected from typical shallow lakes. Log Kow of PBDEs were referenced from the U.S. EPA.

PBDE	BSAF	BCF	log Kow	TMF	*p*	r	Slope
BDE-28	5.51	2.30 × 10^5^	5.94	1.65	0.000	0.62	0.5
BDE-47	3.74	2.32 × 10^5^	6.81	1.63	0.004	0.48	0.49
BDE-100	2.76	1.72 × 10^5^	7.24	1.32	0.049	0.67	0.28
BDE-99	1.14	4.72 × 10^4^	7.32	1.09	0.077	0.49	0.09
BDE-153	6.56	4.07 × 10^4^	7.90	1.40	0.069	0.34	0.34
BDE-154	4.28	3.26 × 10^4^	7.82	0.90	0.283	0.50	−0.11
BDE-183	1.67	1.77 × 10^4^	8.27	1.57	0.003	0.06	0.45
BDE-209	1.98	2.39 × 10^5^	6.27	1.49	0.001	0.40	0.4
ΣPBDEs	3.64	7.37 × 10^4^	-	1.54	0.001	0.50	0.43

## Data Availability

The data presented in this study are available on request from the corresponding author.

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
