# Peer review of "Bioaccumulation Behavior and Human Health Risk of Polybrominated Diphenyl Ethers in a Freshwater Food Web of Typical Shallow Lake, Yangtze River Delta"

_ijerph, 2023, doi:10.3390/ijerph20032671_

Round 1
Reviewer 1 Report
This paper addresses an important topic, and the authors present a good volume of interesting data. The focus is novel and timely. Title and abstract actually reflect what is covered in the paper. Language and phrasing is clear and unambiguous to avoid confusion.
The paper is generally well written, and the framework is well laid out. There are some issues listed below that need to be solved: Line 33: The term TMF - A previous explanation of the written abbreviation is necessary, because it is mentioned here for the first time.Line 28: [37]found - required spacing between parentheses and found
Table 1, where not all data is visible, needs to be corrected.
The Figure 4 is quite vague, the letters are poorly visible, there are too many lines inside the image, while the explanations of the terms on the abscissa and ordinate are written in too large a font.
Reviewer 2 Report
This study evaluated the bioaccumulation behavior of PBDEs in aquatic organisms in typical shallow lakes of the Yangtze River Delta and their effects on human health. After reviewing the manuscript, it is found that there were some serious problems in the article. Therefore, it is rejected. Suggestions are provided as follows:
1. In “1. Introduction”, the widespread distribution of PBDEs has been described at great length. However, the authors’ descriptions of ecosystem and human health hazards are not specific enough to adequately support the significance of the study. It is recommended that the author reorganize this part.
2. In “1. Introduction”, the discussion of the research area was too complicated. It is suggested to simplify this section and fully discuss the necessity of selecting this area as the research area to enhance the logic of the paper.
3. The author’s summary of previous studies was too simple to fully support the innovation and necessity of this study. For example, there was no specific summary of previous research methods, so it is suggested to supplement this part.
4. Please fully explain the considerations behind the selection of the sample in this paper. Was collinearity of measurements between different species considered?
5. The aim of this study was to evaluate the bioaccumulation behavior and health risks of PBDEs in freshwater food webs in the study area. Freshwater food webs are an important part of the study, but the relationships between different species within them are not adequately described.
6.The logic of the paper was not clear enough, so it is suggested that the author should separate the discussion from the research results, and summarize the discussion part separately.
7. This study did not fully summarize the shortcomings of this study, nor did it look forward to the future research direction, which made this paper less scientific. It is suggested to add.
8. The summary of the research conclusion was not perfect enough to summarize the research results and significance of this paper, so it is suggested to supplement.
I hope those suggestions can make this paper more scientific and rigorous.
Reviewer 3 Report
In this study, the authors determined the concentrations of Polybrominated diphenyl ethers by means of GC/MS analysis in a total of 154 samples of wild aquatic organism and 22 environmental samples collected from typical shallow lakes located in the Yangtze River. This is a well-written work that deserves to be published. Some minor suggestions are as follows:
1. Explain all abbreviations the first time used both in the Abstract and in the main text. For example: GC/MS.
2. Abstract:
a) Provide the method of health effect evaluation.
b) Provide the full term for GC/MS, TMF, and ΣPBDE.
c) Correct some grammatical errors: “…were indicated that…”.
3. Introduction:
a) Lines 61-62: Provide more details about the presence of PDBEs in the environment like concentrations.
b) No data or information in a quantified way is included in the literature review provided.
4. Materials and Methods
a) Provide details including the purity of all compounds used.
b) There is no reference at all to GC/MS analysis in this section. Make a reference to S1: Experiment analysis. What was the detection limit?
Round 2
Reviewer 2 Report
I like to thank the authors as they both improved the manuscript after taking all my suggestions into consideration and provided me with a satisfying reply (in the response letter). Hence, I'm pleased to recommend the editor to accept manuscript's current version for publication.
